# Effect of Rhizospheric Fungus on Biological Control of Root Rot (*Fusarium equiseti*) Disease of *Saposhnikovia divaricata*

Zhongming Han [1] , Yi Cui [1], Yan Wang [1], Yunhe Wang [1], Zhuo Sun [1,2,*], Mei Han [1] and Limin Yang [1,*]

[1] College of Chinese Medicinal Materials, Jilin Agricultural University/Cultivation Base of State Key Laboratory for Ecological Restoration and Ecosystem Management of Jilin Province, Changchun 130118, China

[2] State Local Joint Engineering Research Center of Ginseng Breeding and Application, Changchun 130118, China

* Correspondence: 329575068@163.com (Z.S.); ylmh777@126.com (L.Y.)

**Abstract:** *Saposhnikovia divaricata* is a high-demand medicinal plant containing various bioactive metabolites (e.g., chromone). However, root rot disease leads to a dramatic reduction in the yield and quality of *S. divaricata*. The use of rhizospheric microorganisms is one of the best strategies for biological control. In this study, a total of 104 fungi isolated from the rhizospheric soil of *S. divaricata* plants were examined for their different antifungal properties. Subsequently, strain MR-57 was selected as a potential stock for biocontrol due to its broad-spectrum antagonistic activity against pathogens, including *F. equiseti*. Based on the analysis of morphological properties and rDNA internal transcribed spacers (ITSs), strain MR-57 was identified as *Acrophialophora jodhpurensis* (GenBank No. OK287150.1), a newly recorded species for China. In an in vitro antifungal assay, the culture filtrate of strain MR-57 significantly reduced the conidial germination rate and induced alterations in the mycelia morphology of *F. equiseti*, such as deformation and degradation. To assess the antifungal efficacy of MR-57 against root rot disease and the properties promoting the growth of *S. divaricata*, pot experiments were performed under natural outdoor conditions. The results indicated that co-inoculation with MR-57 delayed the occurrence of *S. divaricata* root rot and showed a control efficacy of 65.41% ($p < 0.05$) based on the measurement of suppressed disease lesions. Additionally, MR-57 successfully colonized and formed a stable population in the soil in which *S. divaricata* was grown, and it exhibited a consistently positive effect on the promotion of the growth of *S. divaricata* plants. In short, *Acr. jodhpurensis* MR-57 could be considered for the development of a potential biocontrol agent for the management of *S. divaricata* root rot caused by *F. equiseti*.

**Keywords:** *Aspergillus magaliesburgensis*; antagonistic fungi; colonization; *Fusarium equiseti*; growth promotion





## 1. Introduction

As a global pathogenic fungus of the genus *Fusarium* with a broad range of hosts, *Fusarium equiseti* can cause disease in more than 100 crop species, including agricultural, horticultural, and medicinal plants [1–3]. *Saposhnikovia divaricata* (Turcz.) Schischk. is one of the most important Chinese herbal medicinal plant resources and is officially listed in the pharmacopeia of the People's Republic of China [4]; it is mainly distributed in Northeast China and Inner Mongolia [5]. As an important raw material for the proprietary Chinese medicine Fangfeng Tongsheng Pills recommended for the diagnosis and treatment of COVID-19 [6], as well as for the export of traditional Chinese medicinal materials, *Saposhnikovia divaricata* (Radix Saposhnikoviae) has been shown to have activities against 'various human diseases and conditions, including analgesic, antioxidant, anti-inflammatory activities [7,8] and anti-proliferative, anti-allergic, and anti-cancer effects [9]; it has a broad application prospect and considerable economic value.

Cultivated areas of *S. divaricata* are increasing in northern China as the amount of available natural *S. divaricata* is rapidly decreasing due to over-diggings, and they have become the main source of medicinal material *S. divaricata* [10,11]. The problem is that root rot disease caused by overwintering spores of *F. equiseti* has become a limiting factor for the cultivation of *S. divaricata*. After an outbreak of the disease, the symptoms of root rot are brown rot spots on the main roots, yellowing of the leaves, and eventually wilting and metabolic failure of the aboveground plant of *S. divaricata* [12]. The annual incidence rate is more than 15% to 20%, which negatively affects the yield and quality of *S. divaricata* [12]. Although synthetic agrochemicals can effectively prevent and control root rot of *S. divaricata*, chemical fungicides continue to be the primary strategy for controlling plant disease. However, excessive fungicide use can result in problems including agroecological pollution, soil microecological imbalance, and resistant pathogens [13]. Consequently, environmentally friendly and sustainable management necessitates the utilization of alternative approaches that can inhibit the development of pathogens [14].

In line with the concept of green and safe plant protection, probiotic microorganisms for plant disease control have the advantages of high efficacy, high ecological safety, and substantial economic benefits [15]. The utilization of biocontrol agents (BCAs) is one of the most effective alternatives for reducing chemical pesticides and improving sustainable agricultural production [16]. Therefore, the cultivation of microbial resources with biological control potential has become an important area of plant protection research [17]. As an important area of material exchange between plant roots and the outside world, rhizospheric soil contains abundant beneficial microbial resources, which typically exhibit spatial and nutritional competition hyperparasitism and antagonism with plant pathogenic microorganisms, enhancing plant disease resistance and promoting plant growth and metabolism. Therefore, researchers are particularly interested in microbial resources [18,19]. It has been proven that probiotic microorganisms such as *Trichoderma* species [20], *Bacillus* species [21], and *Pseudomonas* species [22] are particularly effective in plant disease control. In addition, *Bacillus subtilis* [23] and *Trichoderma harzianum* [24,25] have been developed as biological agents for widespread use in controlling plant diseases. However, biological control of fungal diseases such as *S. divaricata* root rot has not been reported in China or elsewhere. Therefore, in this study, the rhizospheric soil of healthy *S. divaricata* was screened for the antagonistic fungal strain MR-57 with significant inhibitory activity against *F. equiseti*. Based on morphological and ITS gene sequence analysis, the taxonomic status of strain MR-57 was established, and the broad-spectrum inhibition, colonization, and biocontrol abilities of MR-57 against root rot caused by *F. equiseti* were systematically evaluated. This study aims to provide a high-quality biocontrol source for the development of biological agents against *S. divaricata* root rot and to lay a theoretical foundation for the biological control of plant diseases.

## 2. Materials and Methods

### 2.1. Microorganisms

A total of 104 fungal isolates were first isolated from rhizospheric soils of *S. divaricata* in the experimental field of Jilin Agricultural University, Changchun, China (43°48′24″ N, 125°24′59″ E, 251 m.a.s.l.), according to the method described by Mirsam et al. [26], and stored to evaluate antagonistic activity against *Fusarium equiseti*. Potato dextrose agar (PDA) and potato dextrose broth (PDB) were prepared for the culture of rhizospheric fungi and plant pathogenic fungi [27]. All pathogens used in this study (*Fusarium equiseti*, *F. oxysporum*, *Alternaria tenuissima*, *A. liriodendron*, *Botrytis cinerea*, *Cylindrocarpon destructans*, *Mycocentrospora acerina*, *Rhizoctonia solani*, *Phytophthora cactorum*) were obtained from the Department of Plant Protection, Jilin Agricultural University in China and were previously isolated from infected plants such as *S. divaricata*, *Schisandra chinensis*, *Asarum sieboldii* Miq., *Glycyrrhiza uralensis* Fisch., and *Panax ginseng* C. A. Meyer. These fungi were maintained on PDA medium and lyophilized filter papers for short-term and long-term storage.

### 2.2. In Vitro Antifungal Activity

Fungal isolates from rhizospheric soil were screened for their antagonistic activity against *F. equiseti* using dual culture assays and confrontation culture assays [28,29], and nine plant pathogenic fungi, namely *Fusarium equiseti*, *Phytophthora cactorum*, *F. oxysporum*, *Alternaria tenuissima*, *A. liriodendron*, *Botrytis cinerea*, *Cylindrocarpon destructans*, *Mycocentrospora acerina*, and *Rhizoctonia solani*, were used as indicators to test the broad-spectrum antifungal activities of the MR-57 isolate using dual culture assays [28].

For the antagonistic activity assays of 104 fungal isolates against *F. equiseti* and antagonistic activity assay of the MR-57 isolate against nine plant pathogenic fungi, the isolates were cultivated as dual cultures on PDA in Petri dishes ($\varnothing$ = 90 mm) at 25 °C in the dark for 7 days, and three replicates were prepared for each assay. PDA dishes inoculated only with the pathogen were used as controls.

### 2.3. Identification of MR-57 Isolate

Observation of culture characteristics: MR-57 isolate was inoculated in the cornmeal agar (CMA), malt extract agar (MEA), oatmeal agar (OA), and plate count agar (PCA) medium plates in three-point fashion, under aseptic conditions, with three replicates each. Plates were incubated at 25 °C in the dark for 10 days, colony diameters were measured, and key properties were used to describe the MR-57 isolate, including colony texture; the abundance, texture, and color of mycelia; and the presence and colors of soluble pigments and exudates [30]. The color of the fungal colony was determined by comparison with the color charts of the International Society Color Council (ISCC) and the National Bureau of Standards (NBS).

Micromorphology observation: Morphological properties of strain MR-57, such as mycelia, conidia, and sclerotia, were observed and recorded using a ZEISS sigma300 field emission scanning electron microscope (Carl Zeiss, Jena, Germany).

Molecular identification: MR-57 isolate was inoculated into PDB and incubated on a rotary shaker (150 rpm) at 25 °C for 5 days. Genomic DNA was extracted from the mycelia of the MR-57 isolate using TaKaRa MiniBEST Universal Genomic DNA Extraction Kit Ver.5.0 (Takara Bio, Shiga, Japan) according to the manufacturer's instructions, and DNA quality was determined by 1% agarose gel electrophoresis [31]. The PCR amplification and sequencing of the MR-57 ITS rDNA was performed using the following primer set: ITS1: 5′-TCCGTAGGTGAACCTGCGG-3′ and ITS4: 5′-TCCTCCGCTTATTGATATGC-3′. The PCR reaction was performed in a 25 μL mixture containing 12.5 μL of 2 × Taq PCR Master Mix (Solarbio, Beijing, China), 1 μL of 10 pmol·L$^{-1}$ of ITS1 and ITS4 primers, 2 μL of genomic DNA, and 8.5 μL of ddH$_2$O. Meanwhile, PCR amplification was performed using a thermal cycler (Thermo Fisher Scientific, Waltham, MA, USA): denaturation at 94 °C for 3 min; annealing at 30 cycles each for 94 °C for 30 s, 55 °C for 30 s, and 72 °C for 1 min; and finally, extension at 72 °C for 5 min. The PCR amplification products were stored at −20 °C and sent to Sangon Biotech (Shanghai, China) for sequencing. The sequence result of the ITS rDNA of the MR-57 isolate was submitted to the NCBI's Nucleotide database (https://blast.ncbi.nlm.nih.gov/Blast.cgi, accessed on 16 February 2022) for comparative analysis using the Basic Local Alignment Search Tool (BLAST). The multi-comparison of Clustal X was performed using the MEGA 5.2 software (https://www.megasoftware.net/megamac.php, accessed on 16 February 2022, Arizona State University, Phoenix, The United States of America), and the phylogenetic tree was constructed using the neighbor-joining (NJ) method.

### 2.4. Antagonistic Activity of MR-57 on Mycelial Growth of F. equiseti

The antagonistic activity of MR-57 culture filtrate was evaluated on a PDA mixture medium using a modified variant of culture filtrate assays (CFAs) described by Millan et al. (2021) [32]. In short, activated mycelia of MR-57 were inoculated into a 250 mL triangular flask containing 100 mL of PDB on a rotary shaker (150 rpm) at 25 °C for 6 days. The broth obtained was centrifuged at 8000 rpm for 15 min, and the supernatant was

filtered through 0.22 μm filters. A sterile PDA mixture medium was obtained by mixing the culture filtrate of MR-57 with PDA in a ratio of 1:4 (*v/v*). The agar–mycelium discs of *F. equiseti* (8 mm diameter) were collected from the edge of a colony of actively growing fungi for CFA, and were placed in the center of PDA media with culture filtrate of MR-57. In the PDA medium without culture filtrate as a control, three replicates were prepared for each. All treatments were cultured in the dark for 5 days at a constant temperature (25 °C), and the mycelial morphology of *F. equiseti* was observed daily. The inhibition rate (%) of MR-57 culture filtrate was determined using the following formula:

$$\text{Inhibition rate (\%)} = (A_c - A_f)/A_c \times 100$$

where $A_c$ refers to the diameter of *F. equiseti* colonies growing in PDA, and $A_f$ refers to the diameter of *F. equiseti* colonies growing in PDA with MR-57 culture filtrate.

### 2.5. Antagonistic Activity of MR-57 on Conidial Germination of F. equiseti

To prepare the spore suspension of the fungal pathogen, *F. equiseti* was incubated on PDA at 25 °C for 10 days. Subsequently, the surface of *F. equiseti* conidia on PDA was eluted with sterile distilled water, and fungal spores were collected using a spreader and filtered through cheesecloth. The spore concentration was adjusted to $1 \times 10^6$ CFU·mL$^{-1}$, and the spores were stored at 4 °C for future use.

The spore suspension of *F. equiseti* was mixed with MR-57 culture filtrate at a ratio of 1:1 (*v/v*) and mixed with sterile distilled water as a control; three replicates were prepared for each. All treatments were incubated at a constant temperature (25 °C). At 6 h, 12 h, 24 h, and 48 h, the conidia of *F. equiseti* were observed using the method described by Zhang et al. (2019) [33].

### 2.6. Soil Colonization Assays

A modified variant of the rifampicin-resistant (Rift) mutant as described by Darma et al. (2020) [34] was utilized. In short, for successive cultures, the Rift mutant of the MR-57 isolate was inoculated into PDB containing increasing concentrations of rifampicin (Rif, Shanghai Macklin Biochemical Co., Ltd., Shanghai, China) at 50, 100, 200, 300, 350, and 400 μg·mL$^{-1}$. The stability of Rift mutants of the MR-57 isolate was tested by subculturing on PDA with 400 μg·mL$^{-1}$ of Rift without significant change compared with the MR-57 isolate, and the Rift mutants of the MR-57 isolate were labeled MR$_{Rif}$-57 and stored at −20 °C.

To prepare spore suspension of strain MR$_{Rif}$-57, the same method as for *F. equiseti* suspension was utilized, and the spore concentration was adjusted to $1 \times 10^7$ CFU·mL$^{-1}$. To evaluate the soil colonization capacity of strain MR-57, a modified variant of the colonization assay described by Chen et al. (2013) [35] was performed. In short, soil samples in which *S. divaricata* was grown were collected from Changchun, China (43°48′16″ N, 125°24′39″ E, 249 m a.s.l.). The spore suspension of MR$_{Rif}$-57 was mixed evenly with the soil in which *S. divaricata* was grown at a ratio of 1:10 (*v/v*) and then placed in a polypropylene pot with a diameter of 28 cm and a height of 18 cm. A seedling of one-year-old *S. divaricata* was planted into each pot, and 20 replicates in a completely randomized block design were prepared for each treatment. In this assay, all treatments were performed in a phytotron for 35 days (16 h of sunlight at 28 °C and 8 h of darkness at 16 °C). After 7 to 35 days of inoculation, strain MR$_{Rif}$-57 was recovered and isolated from the soil, and the amount of soil colonization was determined and recorded.

### 2.7. Biocontrol of MR-57 Isolate

A modified variant of the antifungal assay described by Gholami et al. (2018) and Huang et al. (2020) [36,37] was performed. In short, to prepare a spore suspension of the MR-57 isolate, the same method as for *F. equiseti* was utilized, and the spore concentration was adjusted to $1 \times 10^7$ CFU·mL$^{-1}$. The soil matrix (soil in which *S. divaricata* was grown, vermiculite; 2:1, *v/v*) had been pre-infected with *F. equiseti.* For the biological control assay,

one-year-old *S. divaricata* plants (collected from the medicinal plant garden of Jilin Agricultural University) were utilized. Assays were conducted for the five different treatments as follows: the untreated control (clear water), the Mancozeb 70% WP (Sichuan Runer Technology Co., Ltd.,Chengdu, China, 0.2 g·L$^{-1}$) fungicide treatment, the bacterial suspension of *Bacillus subtilis* (Shandong Lukang Biological Pesticide Co., Ltd., 10$^8$ CFU·mL$^{-1}$), the spore suspension of *Trichoderma harzianum* (Shandong Lvlong Biological Co., Ltd., Weifang, China, $1 \times 10^7$ CFU·mL$^{-1}$), and the spore suspension of the MR-57 isolate ($1 \times 10^7$ CFU·mL$^{-1}$). Twenty replicates were prepared for each treatment in a completely randomized block design. After 70 days of inoculation with *F. equiseti*, the disease index of *S. divaricata* root rot and the root rot disease control efficacy were calculated.

The disease rating scale (0–9) for root rot was as follows: 0 = no lesion on the roots, healthy; 1 = lesions covering < 10% of the roots; 3 = lesions covering 11% to 25% of the roots; 5 = lesions covering 26% to 50% of the roots; 7 = lesions covering 51% to 75% of the roots; 9 = lesions covering 76% to 100% of the roots, seedling is dead.

$$\text{Disease index (DI)} = [(0N_0 + 1N_1 + 3N_3 + 5N_5 + 7N_7 + 9N_9)/9N] \times 100,$$

$$\text{Control efficacy (CE, \%)} = [DI_{(CK)} - DI_{(treatment)}]/DSI_{(CK)} \times 100\%,$$

where $N_0$ to $N_9$ represent the numbers of plants with each corresponding disease scale and N represents the total number of plants assessed.

### 2.8. Plant Growth Promotion of MR-57 Isolate

To evaluate the influence of the spore suspension of MR-57 on the yield of *S. divaricata*, plants were grown in a polypropylene pot with a diameter of 28 cm and a height of 20 cm filled with soil in which *S. divaricata* was grown and vermiculite (2:1). A seedling of one-year-old *S. divaricata* (collected from the medicinal plant garden of Jilin Agricultural University) was transplanted into each pot, and 20 replicates in a completely randomized block design were prepared for each treatment. Assays were conducted for the four different treatments as follows: the untreated control (water), the bacterial suspension of *Bacillus subtilis* (10$^8$ CFU·mL$^{-1}$), the spore suspension of *Trichoderma harzianum* ($1 \times 10^7$ CFU·mL$^{-1}$), and the spore suspension of the MR-57 isolate ($1 \times 10^7$ CFU·mL$^{-1}$). After 60 days, for each treatment, nine *S. divaricata* plants were randomly selected, and stem length from the soil line to the top was measured. Subsequently, the *S. divaricata* plants were carefully uprooted; washed under running tap water; and individually measured for the growth parameters of *S. divaricata*, including root length, plant fresh weight, and root fresh weight. *S. divaricata* plants were subsequently dried overnight at 50 °C, and the dry weights of the whole plant and root were measured.

### 2.9. Statistical Analysis

The assays were repeated at least three times with three replications in each repetition. The data presented for each assay are the means (±standard deviation) of three experiments.

The results were analyzed using one-way ANOVA with 95% confidence intervals for Duncan's DMRT. All statistical analyses were performed using SPSS Statistics 13.0 (SPSS Inc., Chicago, IL, USA) and graphed with OriginPro 9.5 (OriginLab Corporation, Northampton, MA, USA).

## 3. Results
### 3.1. Antagonistic Activities of Fungal Isolates against F. equiseti

In this study, the in vitro antagonistic activities of 104 fungal isolates were evaluated against *F. equiseti*, which causes root rot in *S. divaricata*. Eight of the isolates (7.55%) displayed distinct growth inhibition against *F. equiseti* on PDA as potential antagonists, with antagonism rates ranging from 55.19% to 60.00%. Compared with the other seven strains of rhizospheric fungi, strain MR-57 exhibited a significantly antagonistic effect against *F. equiseti* ($p < 0.05$), with an inhibition rate of 60.00% and an inhibition zone of

9.05 mm, thereby effectively controlling the spread of the pathogenic fungus (Table S1). According to multiple experimental validations, the inhibition activity of the strain MR-57 was found to be stable, making it a suitable candidate for further research.

### 3.2. Antifungal Spectrum of MR-57 Isolate

The fungal isolate MR-57 showed the capability to inhibit the growth of nine common plant pathogenic fungi at rates ranging from 59.25% to 75.18% (Figure 1), four of which were strongly inhibited with an inhibition rate over 70%, namely *C. destructans*, *M. acerina*, *A. liriodendron*, and *A. tenuissima*. However, the inhibition rates of strain MR-57 against *F. oxysporum*, *R. solani* were relatively low, averaging 59.44%. The results indicated that the MR-57 isolate possessed broad-spectrum inhibitory ability and exhibited biocontrol potential as a biological control agent for the fungal disease affecting *S. divaricata* and other plants.

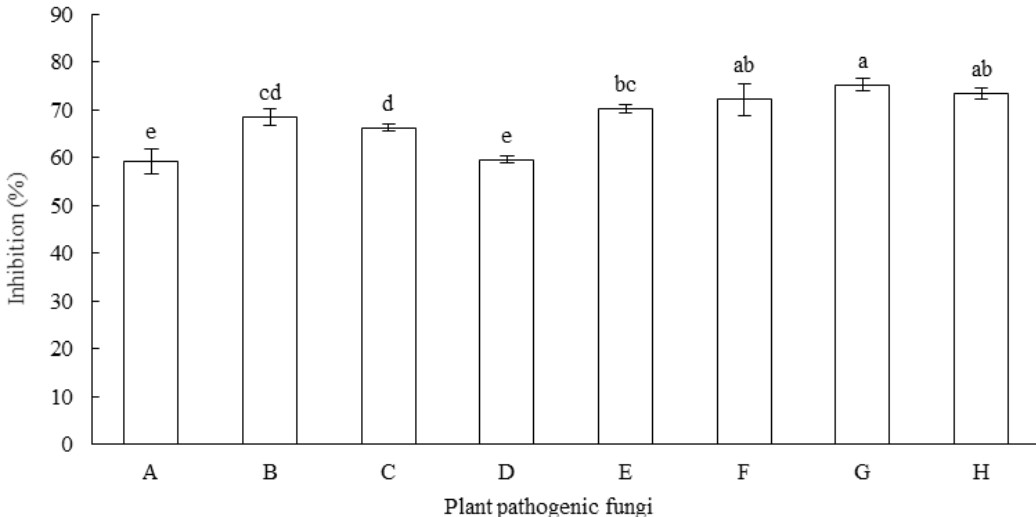

**Figure 1.** Antagonistic activities of MR-57 isolate against fungal pathogens. Growth inhibition percentage (%) of fungal pathogens based on dual culture assays. A: *R. solani*, B: *P. cactorum*, C: *B. cinerea*, D: *F. oxysporum*, E: *A. tenuissima*, F: *A. liriodendron*, G: *C. destructans*, H: *M. acerina*. Vertical bars represent standard deviation of the means ($n = 3$). Bars followed by the different letters are significantly different at $p < 0.05$ according to Duncan's DMRT. *F* value showed 34.596, *p* value showed 0.0001.

### 3.3. Identification of Antagonistic Strain MR-57

#### 3.3.1. Identification of Culture Characteristics

Figure 2 shows the colony morphology of strain MR-57 on CMA, MEA, OA, and PCA media. On CMA, the colonies had a diameter between 28 and 34 mm, a white and flocculent center, an irregular margin with a white border of 2 to 3 mm, and a stale grey front. On MEA, the colonies had a diameter between 24 and 34 mm; a color progression of white, stale grey, and black; and an irregular margin. On OA, the colonies had a diameter between 27 and 30 mm, scattered and sparse mycelia, and a stale grey front with aerial hypha. On PCA, the colonies had a diameter between 23 and 26 mm, a fluffy center, and a white front with aerial hypha (Figure 2).

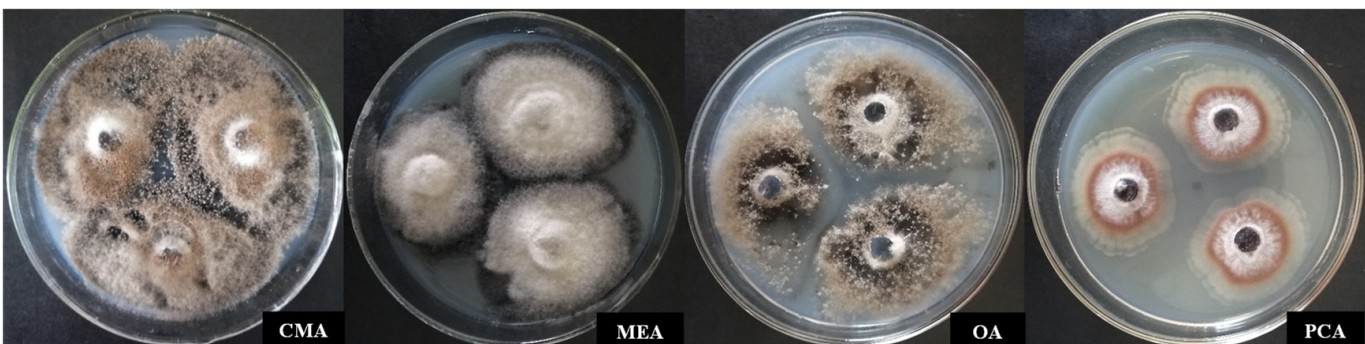

**Figure 2.** Characteristics of the colonies on media and the morphology of MR-57 isolate. The front of the colony was cultured on CMA, MEA, OA, and PCA at 25 °C in the dark for 10 days.

### 3.3.2. Identification of Microscopic Features

Ascomata were superficial, subglobose to ovate, between 120 and 200 µm high, between 80 or 120 and 160 µm in diameter, and consisted of intricate texture on the surface of the ascomatal wall. Lateral hairs were flexuous. Ascospores were (4.5−) 6 − 7.0 (−7.5) µm × (11.0−) 1214.0 (−15) µm, rough, and ellipsoidal. No conidial morph was observed (Figure 3). All these features were similar to those of the species in genus *Acrophialophora* of Chaetomiaceae as described by Wang et al. (2019) [38].

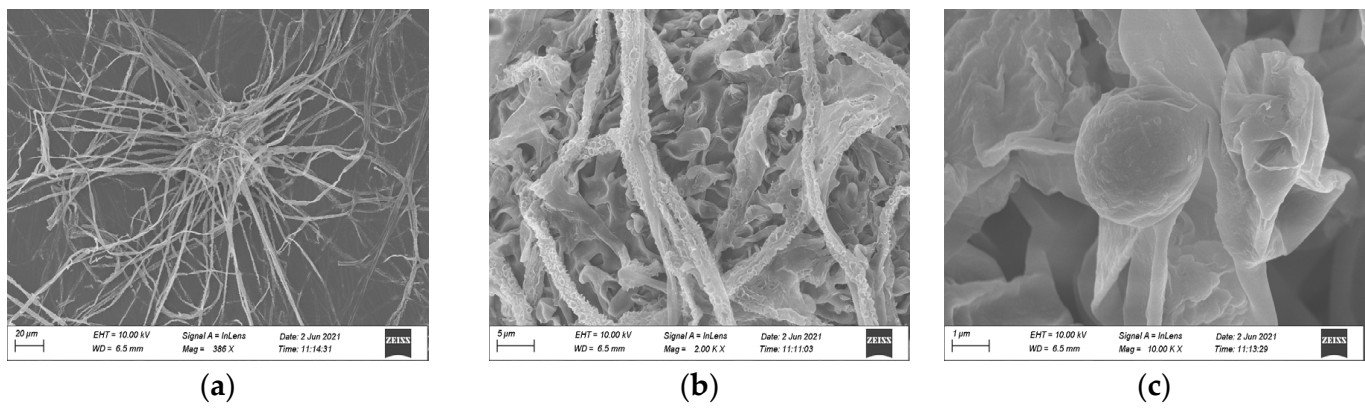

**Figure 3.** The reproductive structures of MR-57 isolate: (**a**) ascomata, (**b**,**c**) ascospores.

### 3.3.3. Sequence and Phylogenetic Analysis

The ITS gene sequence of strain MR-57 was amplified by PCR and sequenced by Sangon Biotech (Shanghai) to yield a base sequence of 482 bp with GenBank accession number OK287150.1. According to the analysis and comparison of 5.8S rRNA gene fragments by NCBI's BLAST, strain MR-57 showed the highest sequence homology (>99%) with *Acrophialophora jodhpurensis* (MN889997.1), *Thielavia hyrcaniae* (MH860334.1), *Chaetomium jodhpurense* (KP336749.1), and *Collariella gracilis* (MH864008.1), and the homology among strain MR-57, *Acr. Levis* (MK336561.1), and *Acr. Ellipsoidea* (LC485179.1) approached 97%. Based on phylogenetic tree construction, strain MR-57 and *Acr. jodhpurensis* (NR_165583.1) had high homology and were in the same clade (Figure 4), which was identified as *Acr. jodhpurensis*.

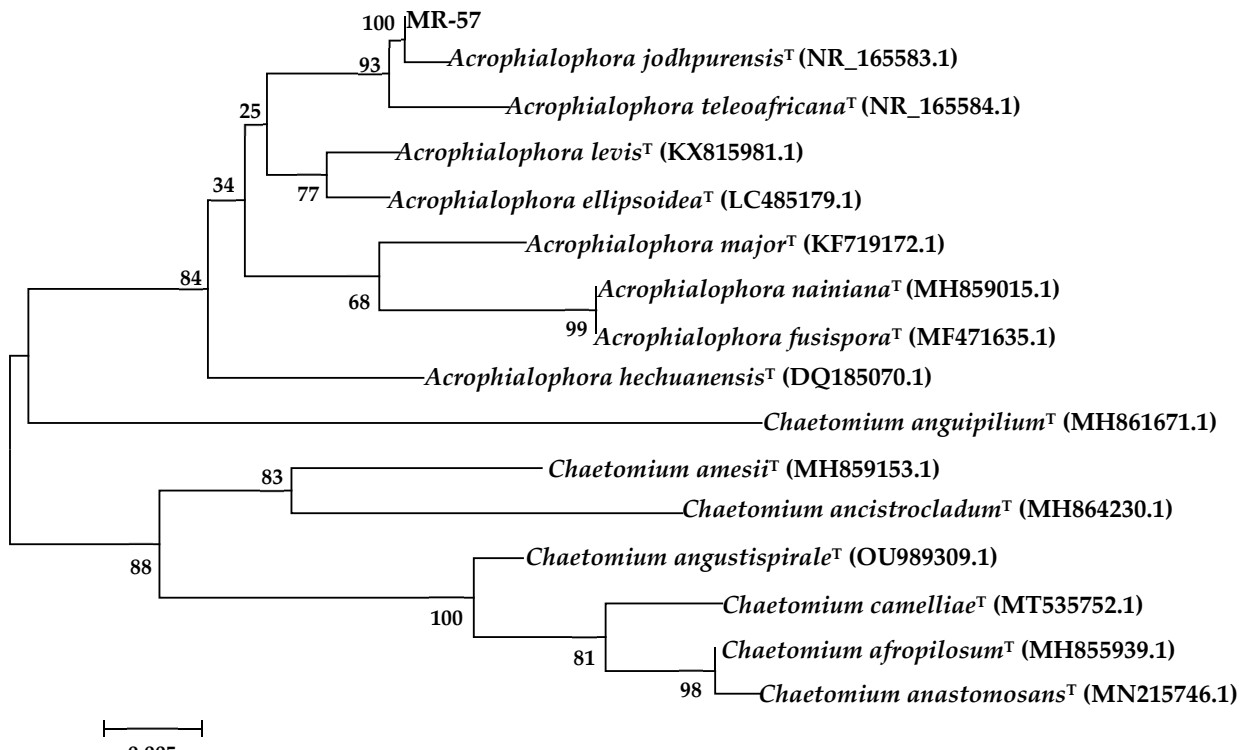

**Figure 4.** Phylogenetic tree of MR-57 isolate and closely related taxa based on ITS sequences, constructed using the neighbor-joining method. The values of significant bootstrap (>43%) and the scale (0.01) of the phylogenetic tree are exhibited.

*3.4. MR-57 Culture Filtrate against Mycelia of F. equiseti*

The MR-57 culture filtrate showed a certain inhibitory effect on the mycelial growth of *F. equiseti*. The inhibited pathogenic colonies became thinner, with distinct edges, sparse aerial mycelium, and a slow growth trend, and the mycelia significantly dissolved (Figure 5a,b). A microscopic observation of the mycelial morphology of *F. equiseti* revealed that healthy mycelia of *F. equiseti* were observed with a smooth exterior surface (Figure 5c). However, the surface of inhibited mycelia was rough and uneven, increased with irregular extension, and the branching ends were enlarged, with condensed inclusions and uneven distribution. The inhibited mycelia were distorted, folding, constricted, broken, and fractured (Figure 5d,e). The results indicated that strain MR-57 could inhibit the growth and development of *F. equiseti* through the production of particular substances.

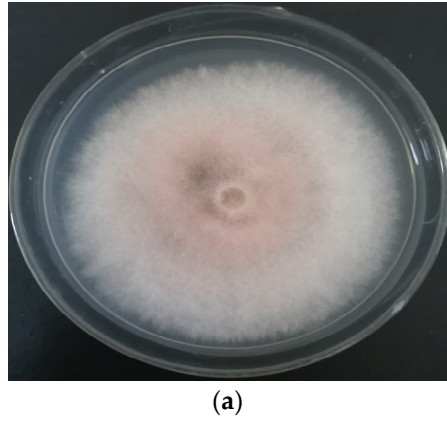

(**a**)

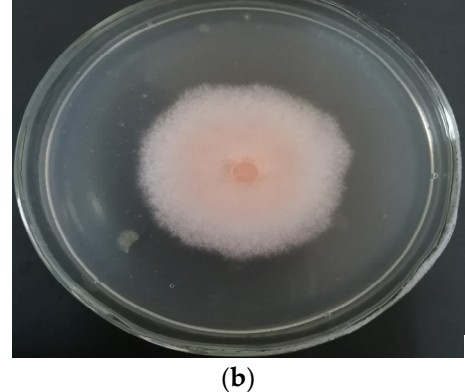

(**b**)

**Figure 5.** *Cont.*

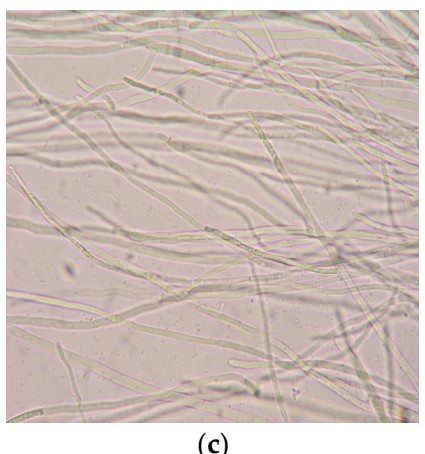
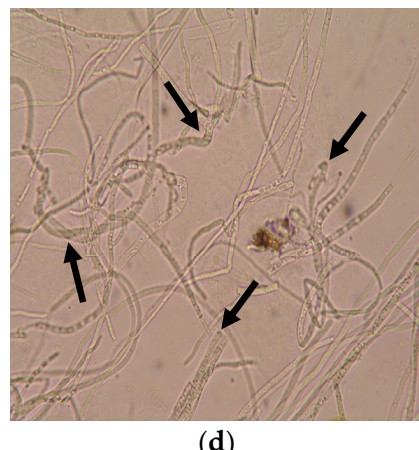
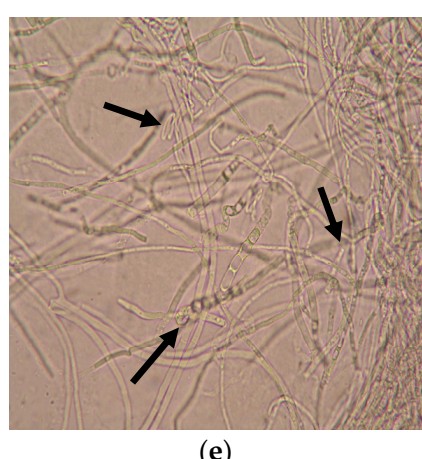

(**c**) (**d**) (**e**)

**Figure 5.** Effect of culture filtrate of *Acr. jodhpurensis* MR-57 on the hyphal morphology of *F. equiseti*. (**a**) PDA media inoculated only with *F. equiseti*. (**b**) Mycelia of *F. equiseti* cultures on PDA in the presence of culture filtrate of *Acr. jodhpurensis* MR-57. (**c**) Healthy mycelia of *F. equiseti*. (**d**,**e**) Microscopic observation of mycelial growths of *F. equiseti* during co-culture assay with the absence or presence of culture filtrate of *Acr. jodhpurensis* MR-57.

*3.5. MR-57 Culture Filtrate Assay on Spore Germination of F. equiseti*

The investigation of the co-culture of the MR-57 culture filtrate and the *F. equiseti* spore suspension indicated that strain MR-57 has a significant inhibition ability against the spore germination of the pathogen. The spore germination rate of *F. equiseti* in co-culture for 48 h was 24.93%, with an inhibition rate reaching 71.01% (Table 1).

**Table 1.** Effects of culture filtrate of MR-57 on the spore germination of *F. equiseti*.

| Treatment | Spore Germination Rate (%) | | | | Inhibition Rate at 48 h (%) |
|---|---|---|---|---|---|
| | **6 h** | **12 h** | **24 h** | **48 h** | |
| Sterile distilled water (CK) | $37.00 \pm 5.91$ [a] | $43.19 \pm 4.45$ [a] | $58.60 \pm 6.40$ [a] | $86.03 \pm 10.24$ [a] | — |
| MR-57 | $3.84 \pm 1.14$ [b] | $8.26 \pm 0.56$ [b] | $22.88 \pm 5.83$ [b] | $24.93 \pm 3.74$ [b] | $71.01 \pm 2.40$ |
| *F* value | 92.905 | 181.365 | 51.051 | 94.194 | |
| *p* value | 0.0006 | 0.0002 | 0.002 | 0.0006 | |

Conidial in vitro germination rate of *F. equiseti* (Mean $\pm$ SD) at 6, 12, 24, and 48 h after inoculation with addition of culture filtrate of MR-57 isolate at 25 °C. Different letters indicate significant differences for the same column at $p < 0.05$ according to Duncan's DMRT.

*3.6. Soil Colonization Ability of MR-57 Isolate*

Strain MR$_{Rif}$-57 was still able to grow stably on PDA plates containing rifampicin with a concentration of 400 µg·mL$^{-1}$ after 10 generations of subculture without a discernible change in the morphology. Strains MR-57 and MR$_{Rif}$-57 were similarly effective against *F. equiseti*, with an inhibition rate of up to 60% for strain MR$_{Rif}$-57. The results indicate that strain MR$_{Rif}$-57 was genetically stable and maintained a high level of inhibition activity against *F. equiseti*. MR$_{Rif}$-57 was tested for its ability to colonize soil for up to 35 days, and it was shown that the population size of MR$_{Rif}$-57 first increased and then decreased in the soil. On the 7th day after inoculation, the MR$_{Rif}$-57 population was $1.80 \times 10^6$ CFU·g$^{-1}$ of soil. On the 21st day after inoculation, the MR$_{Rif}$-57 population was $2.83 \times 10^6$ CFU·g$^{-1}$ of soil, reaching its peak. Subsequently, the population abundance began to decrease. On the 35th day after inoculation, the MR$_{Rif}$-57 population eventually stabilized at $1.92 \times 10^6$ CFU·g$^{-1}$ of soil (Figure 6).

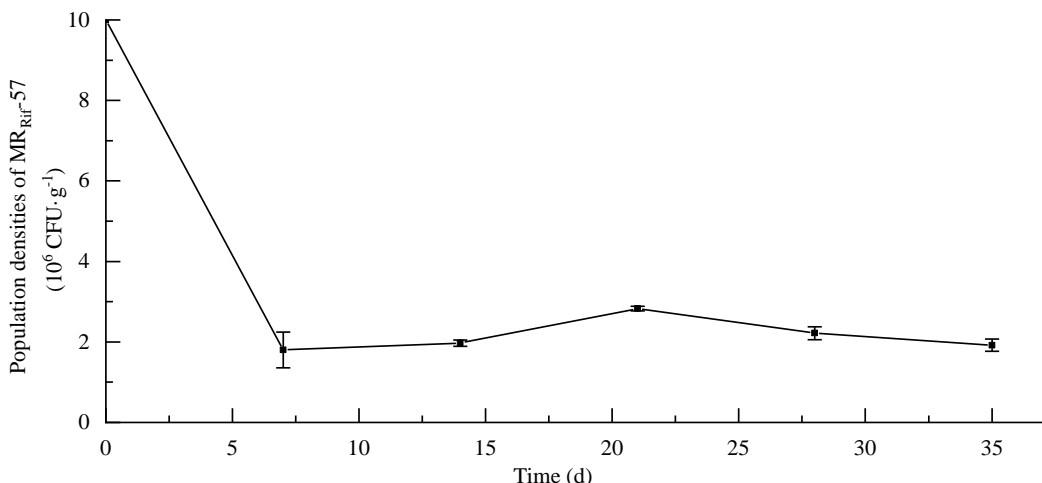

**Figure 6.** Time course of MR$_{Rif}$-57 colonization in the soil in which *S. divaricata* was grown. Soil samples were treated with a spore suspension of MR$_{Rif}$-57 at a final density of $10^7$ CFU·g$^{-1}$ of soil. After 7, 14, 21, 28, and 35 days of inoculation, the population densities of the MR$_{Rif}$-57 were determined.

### 3.7. Biocontrol Efficiency of MR-57 against Root Rot of S. divaricata

To evaluate the efficacy of the MR-57 isolate in controlling root rot of *S. divaricata*, the *S. divaricata* plants were planted in a soil matrix infected with *F. equiseti*. There were fewer disease symptoms in *S. divaricata* plants treated with a spore suspension of MR-57 or Mancozeb 70% WP (Table 2). Ten weeks after treatment, in the untreated (water) control tested, the *S. divaricata* plants inoculated with *F. equiseti* showed large-scale disease spots on the leaves with a disease severity index of 64.14, whereas treatment with MR-57 significantly decreased root rot of *S. divaricata* with antifungal efficacy levels of 65.41% for the spore suspension of MR-57. In addition, the antifungal efficacy with a spore suspension of MR-57 was significantly different ($p < 0.05$) from that of the bacterial suspension of *B. subtilis* (49.99%) and the spore suspension of *T. harzianum* (51.89%), but it was not significantly distinct from that of Mancozeb 70% WP (63.44%).

**Table 2.** Effect of the MR-57 spore suspension on root rot of *S. divaricata*.

| Treatment | Disease Index | Disease Control (%) |
|---|---|---|
| MR-57 | 22.18 ± 3.72 [c] | 65.41 ± 5.80 [a] |
| Mancozeb 70% WP | 23.44 ± 2.11 [c] | 63.44 ± 3.30 [a] |
| *B. subtilis* | 32.07 ± 5.66 [b] | 49.99 ± 8.82 [b] |
| *T. harzianum* | 30.85 ± 4.29 [b] | 51.89 ± 6.70 [b] |
| Clear water (CK) | 64.14 ± 2.18 [a] | — |
| *F* value | 59.701 | 4.434 |
| *p* value | 0.0001 | 0.0409 |

Different letters indicate significant differences for the same column at $p < 0.05$ according to Duncan's DMRT.

### 3.8. Effects of MR-57 on Plant Growth Characteristics of S. divaricata

As shown in Table 3, the MR-57 isolate, *T. harzianum*, and *B. subtilis* had a significant impact on the growth characteristics of *S. divaricata* plants. Seedling fresh weight, seedling dry weight, root fresh weight, and root dry weight of *S. divaricata* treated with strain MR-57 were significantly greater than those of other treatment groups ($p < 0.05$). Compared with the CK treatment group (clear water), the fresh weight of *S. divaricata* increased by 112.42% in the MR-57 treatment group, and the spore suspension of MR-57 promoted the indicators of *S. divaricata* plants, including seedling dry weight, root fresh weight, and root dry weight, which increased by more than 104.98%. This demonstrated that the MR-57 isolate could stimulate the growth of *S. divaricata* plant as a plant growth-promoting fungus (PGPF), and it has potential as a microorganism beneficial for yield increase.

**Table 3.** Effects of MR-57 on promoting growth of *S. divaricata*.

| Treatment | Stem Length (cm) | Root Length (cm) | Seedling Fresh Weight (g) | Root Fresh Weight (g) | Seedling Dry Weight (g) | Root Dry Weight (g) |
|---|---|---|---|---|---|---|
| MR-57 | $27.71 \pm 3.22$ [ns] | $30.36 \pm 2.88$ [ns] | $20.18 \pm 3.09$ [a] | $6.64 \pm 0.81$ [a] | $4.94 \pm 0.77$ [a] | $1.43 \pm 0.03$ [a] |
| *B. subtilis* | $24.49 \pm 4.38$ [ns] | $29.14 \pm 2.76$ [ns] | $10.44 \pm 0.80$ [b] | $4.10 \pm 0.71$ [b] | $2.56 \pm 0.69$ [b] | $0.85 \pm 0.07$ [b] |
| *T. harzianum* | $24.32 \pm 5.53$ [ns] | $28.03 \pm 3.83$ [ns] | $11.97 \pm 1.22$ [b] | $4.16 \pm 0.28$ [b] | $3.19 \pm 0.12$ [b] | $0.90 \pm 0.25$ [b] |
| Clear water (CK) | $24.31 \pm 5.05$ [ns] | $27.96 \pm 2.33$ [ns] | $9.50 \pm 2.48$ [b] | $2.66 \pm 0.15$ [c] | $2.41 \pm 1.29$ [b] | $0.56 \pm 0.15$ [c] |
| *F* value | 0.391 | 0.426 | 16.013 | 25.963 | 5.857 | 17.842 |
| *p* value | 0.7626 | 0.74 | 0.001 | 0.0002 | 0.0204 | 0.0007 |

Different letters indicate significant differences for the same column at $p < 0.05$ according to Duncan's DMRT. ns indicates no significant differences.

## 4. Discussion

In the investigation of beneficial microbial resources in the cultivation base of *S. divaricata* in Changchun, Jilin Province, this study identified a rhizospheric fungus, strain MR-57, with a strong antagonistic effect against *F. equiseti,* which causes root rot disease of *S. divaricata*, that is effective in inhibiting the growth of at least nine common plant pathogenic fungi. Based upon the results of culture characteristics, microscopic features, and molecular identifications, strain MR-57 isolated from the rhizospheric soil of *S. divaricata* was identified as *Acr. jodhpurensis*. For the first time in China, this strain was discovered and reported, and this fungal species was isolated [38,39]. *Acrophialophora jodhpurensis* is a species of genus *Acrophialophora* of Chaetomiaceae, first reported to be isolated from rabbit droppings in Jodhpur, Rajasthan, India [40]. In addition, certain members of the genus *Acrophialophora* have strong ecological competitiveness, and they demonstrated exceptional effects in plant disease biocontrol [41,42], plant resistance induction [43], soil improvement and restoration [44], heavy metal biosorption [45], and other performances. They are valuable sources of biocontrol fungi with a high potential for development and application.

Antagonistic fungi mainly inhibit plant diseases through competition, lysis, and hyperparasitism [46]. Through the production of β-glucosidase, cellulose, xylanase [47,48], pectolases [49], and β-mannanase [50], the genus *Acrophialophora* was discovered to effectively inhibit the growth of plant pathogenic fungi. In this study, MR-57 culture filtrate inhibited *F. equiseti* mycelium and spore germination, resulting in mycelium malformation and rupture and a decreased spore germination rate. It was hypothesized that *Acr. jodhpurensis* MR-57 could destroy the cell wall or membrane structure of *F. equiseti* mycelium by secreting extracellular fungistatic substances. At the same time, the respiration of pathogen spores was disrupted, which inhibited the growth of pathogenic fungi. This suggests that the extracellular secondary metabolites of *Acr. jodhpurensis* have the potential to be developed as biocontrol agents for plant diseases. However, the specific substances responsible for the inhibitory effect of *Acr. jodhpurensis* remain unknown, and further research is still required.

The ability of antagonistic microorganisms to colonize the soil directly affects the efficacy and stability of their biocontrol effects and is also an important indicator for assessing the potential of biocontrol sources [51]. In this study, *Acr. jodhpurensis* MR-57 was proven to possess soil colonization capacity through antibiotic labeling, and it grew and propagated stably in the soil. In future research, the colonization effect of MR-57 on *S. divaricata* will be evaluated.

*Acrophialophora* has been reported to prevent and control plant diseases. Daoodi et al. [52] reported the endophytic fungus *Acr. jodhpurensis* isolated from tomato roots as a potential biocontrol source against *Rhizoctonia solani*. In this study, *Acr. jodhpurensis* was isolated from the rhizospheric soil of *S. divaricata*, suggesting that *Acr. jodhpurensis* may be a member of the soil saprophytic group of fungi (first isolated from rabbit droppings and having saprophytic characteristics), while this species has a strong endophytic ability. In addition, it was demonstrated that *Acr. jodhpurensis* MR-57 could effectively prevent and control root rot of *S. divaricata* and has a growth-promoting effect on *S. divaricata* plants better than that

registered for *B. subtilis* and *T. harzianum*. To our knowledge, this is the first report on the biocontrol activities of *Acr. jodhpurensis* against fungal pathogens of medicinal plants and its plant growth promotion. Large numbers of exogenous microorganisms introduced into the soil are susceptible to competition from dominant habitant microorganisms or other environmental factors that limit their growth and reproduction, resulting in unsatisfactory biocontrol effects. Furthermore, it has been found that non-indigenous microorganisms may have a substitution effect on the dominant habitant microorganisms in the soil microenvironment, thereby altering the soil's original microecological balance [53]. In this study, *Bacillus subtilis* and *T. harzianum*, as exogenous microorganisms not originating from the inherent microecological balance system of the *S. divaricata* cultivation soil, may have competed with indigenous microorganisms for ecological niches after their introduction into the soil, resulting in weakened biological activity and decreased disease control and plant growth promotion capacity.

## 5. Conclusions

*Acrophialophora jodhpurensis* MR-57 was isolated and identified from the rhizospheric soil of *S. divaricata* as a potential biocontrol agent for *S. divaricata*. However, to accurately evaluate the biocontrol efficacy of *Acr. jodhpurensis* MR-57, we suggest conducting additional studies utilizing large-scale and complex *S. divaricata* field conditions.

**Supplementary Materials:** The following supporting information can be downloaded at: https://www.mdpi.com/article/10.3390/agronomy12112906/s1. Table S1. T Antifungal activities of selected rhizospheric fungi against *F. equiseti.*

**Author Contributions:** Conceptualization, Z.S. and L.Y.; methodology, Y.W. (Yan Wang) and Y.C.; software, Z.H.; validation, Y.C. and Z.H.; formal analysis, Z.S. and Z.H.; investigation, Y.W. (Yunhe Wang) and M.H.; resources, Y.W. (Yan Wang) and Y.C.; data curation, Z.H. and Z.S.; writing—original draft preparation, Z.H. and Z.S.; writing—review and editing, Y.W. (Yan Wang) and Y.C.; supervision, Z.H. and Z.S.; project administration, Z.H., Z.S. and L.Y. All authors have read and agreed to the published version of the manuscript.

**Funding:** This research was funded by the National Key Research and Development Project of China (grant numbers: 2019YFC1710700, 2019YFC1710702), Jilin Scientific and Technological Development Program (grant numbers: 20200402104NC, 20200404010YY, 20210204011YY), Promotion Demonstration Project of Forestry Science and Technology of China (grant number: JLT2021-03), and Agriculture Research System of China (grant number: CARS-21).

**Conflicts of Interest:** The authors declare that they have no known competing financial interest or personal relationships that could have appeared to influence the work reported in this paper.

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
