# Peer review of "Effect of Rhizospheric Fungus on Biological Control of Root Rot (Fusarium equiseti) Disease of Saposhnikovia divaricata"

_agronomy, doi:10.3390/agronomy12112906_

Round 1

Reviewer 1 Report (Previous Reviewer 1)

The authors have addressed all the important major mistakes indicated. However, I have some minor suggestions concerning the new changes carried out. Authors should add F value and p value in the results description of figure 1. In table 2, one p value is 0, this is not possible, please they should change by “< than one number”. In table 3, stem length and root length the results are not statistically significant; then, the letter “a” should be substituted by ns (not significant differences) and indicate ns: not significant differences in the table legend. In addition, the authors indicate p value and P value in the manuscript, they should use one terminology for homogenizing the manuscript. Title of Figure 6 should be rewritten: Figure 6. Time course of MRRif-57 colonization in the soil….

Finally, the manuscript has improved from the previous revision, but these minor concerns need to be modified.

Author Response

Dear Reviewer:

Thank you for your letter and for the reviewer’ comments concerning our manuscript entitled “Effect of rhizospheric fungus on biological control of root rot (Fusarium equiseti) disease of Saposhnikovia divaricata”. Those comments are all valuable and very helpful for revising and improving our paper, as well as the important guiding significance to our researches. We have studied comments carefully and we hope meet with approval. All changes have been marked in red in the manuscript. The main corrections in the paper and the responds to the reviewer’s comments are as flowing:

Comment 1: Authors should add F value and p value in the results description of figure 1.

Response: Considering the reviewer’s suggestion, F value and p value been supplemented in the description of figure 1.

Comment 2: In table 2, one p value is 0, this is not possible, please they should change by “< than one number”

Response: Thanks for the reviewer’s suggestion, the p value of table 2 been corrected.

Comment 3: In table 3, stem length and root length the results are not statistically significant; then, the letter “a” should be substituted by ns (not significant differences) and indicate ns: not significant differences in the table legend

Response: Thanks for the reviewer’s suggestion, this part have been corrected in the table 3.

Comment 4: The authors indicate p value and P value in the manuscript, they should use one terminology for homogenizing the manuscript.

Response: Considering the reviewer’s suggestion, the expression of p value have been corrected uniformly in the manuscript.

Comment 5: Title of Figure 6 should be rewritten: Figure 6. Time course of MRRif-57 colonization in the soil….

Response: Thanks for the reviewer’s suggestion. The title of figure 6 have been corrected as “Time course of MRRif-57 colonization in the soil in which S. divaricata was grown”.

We hope that all these changes fulfil the requirements to make the manuscript acceptable for publication in Agronomy.

We deeply appreciate your consideration of our manuscript. If you have any queries, please don’t hesitate to contact me at the address below.

Thank you and best regards.

Yours sincerely,

Zhongming Han

Reviewer 2 Report (Previous Reviewer 2)

All the comments I made in the previous review have been taken into account. Therefore, I recommend the article for publication.

Author Response

Dear Reviewer,

Many thanks for your friendly comment.

Regards,

Zhongming

Reviewer 3 Report (New Reviewer)

The interest is in the use of Acrophialophora jodhpurensis to control Fusarium equiseti in Saposhnikovia divaricata. There are some minor issues that should be addressed prior to the publication of the manuscript.

-Two background sentences in abstract are two long

-Section 2.8 how to measure plant parameters, please give more details

-Line 268, it should be Figure 1, and also repeated in last sentence

-Table 1: CK should have a full form, and make it uniform to table 2

-Lines 343-345: some phrases were repeated

-Line 373, clear water, please check in table 2

-Name of fungi should be full form (line 438)

-Should conclude what we do, so “and other plants” should delete

Author Response

Dear Reviewer,

Thank you for your letter and for the reviewer’ comments concerning our manuscript entitled “Effect of rhizospheric fungus on biological control of root rot (Fusarium equiseti) disease of Saposhnikovia divaricata”. Those comments are all valuable and very helpful for revising and improving our paper, as well as the important guiding significance to our researches. We have studied comments carefully and we hope meet with approval. All changes have been marked in red in the manuscript. The main corrections in the paper and the responds to the reviewer’s comments are as flowing:

Comment 1: Two background sentences in abstract are two long

Response: Considering the reviewer’s suggestion, this part have been corrected as following,

 Saposhnikovia divaricata is a high-demand medicinal plant containing various bioactive metabolites (e.g., chromone). However, root rot disease leads to a dramatic reduction in the yield and quality of S. divaricata.

Comment 2: Section 2.8 how to measure plant parameters, please give more details

Response: Considering the reviewer’s suggestion, the section 2.8 have been reworded as following,

After 60 days, for each treatment, nine S. divaricata plants were randomly selected and stem length from soil line to top was measured. Subsequently, the S. divaricata plants were carefully up-rooted and washed under running tap water, and individually measured for the growth parameters of S. divaricata, including root length, plant fresh weight, root fresh weight. S. divaricata plants were subsequently dried overnight at 50 °C and dry weight of whole plant and root were measured.

Comment 3: Line 268, it should be Figure 1, and also repeated in last sentence

Response: In the section 3.3.1, we described colony morphology of strain MR-57 on different culture medium. So we think that the Figure 1 should be corrected as Figure 2.

Comment 4: Table 1: CK should have a full form, and make it uniform to table 2

Response: Considering the reviewer’s suggestion, the part of table 1 and table 2 have been corrected.

Comment 5: Lines 343-345: some phrases were repeated

Response: Considering the reviewer’s suggestion, this part have been reworded as following,

On the 7th day after inoculation, the MRRif-57 population was 1.80 × 106 CFU·g-1 of soil. At 21th day after inoculation, the MRRif-57 population in the soil achieved a peak, the cell counts showed an increase from 1.80 × 106 CFU·g-1 to 2.83 × 106 CFU·g-1 of soil. 

Comment 6: Line 373, clear water, please check in table 2

Response: Considering the reviewer’s suggestion, this part have been corrected.

Comment 7: Name of fungi should be full form (line 438)

Response: Considering the reviewer’s suggestion, the name of strain MR-57 have been corrected.

Comment 8: Should conclude what we do, so “and other plants” should delete

Response: Thanks. “and other plants” was deleted.

We hope that all these changes fulfil the requirements to make the manuscript acceptable for publication in Agronomy.

We deeply appreciate your consideration of our manuscript. If you have any queries, please don’t hesitate to contact me at the address below.

Thank you and best regards.

Yours sincerely,

Zhongming Han

This manuscript is a resubmission of an earlier submission. The following is a list of the peer review reports and author responses from that submission.

Round 1

Reviewer 1 Report

Manuscript ID: agronomy-1919858

Manuscript comments:

In the present manuscript entitled “New observation in biocontrol activity of Acrophialophora jodhpurensis against Fusarium equiseti on Saposhnikovia divaricata and as a plant growth promoter”, the authors have obtained and evaluated one biological control agent against the disease caused by F. equiseti on S. divaricate. The manuscript is not well structured and presents several deficiencies in its composition. Important comments are indicated below, more major and minor suggestions are highlighted with comments in the attached PDF file.

Abstract

This part should contain brief information of the introduction such as the importance of this crop and disease.

Introduction

This section lacks citations-references, only 10 different references is not acceptable for an introduction in a scientific publication.

Results

The authors should include statistical parameters from analysis of variance (ANOVA) such as F value, dr or p value.

Table 2: A table only with two columns is not acceptable in a scientific publication, authors should change by a figure.

Author Response

Thank you for your letter and for the reviewers’ comments concerning our manuscript entitled “New observation in biocontrol activity of Acrophialophora jodhpurensis against Fusarium equiseti on Saposhnikovia divaricata and as a plant growth promoter” (Manuscript ID: agronomy-1919858). Those comments are all valuable and very helpful for revising and improving our paper, as well as the important guiding significance to our researches. We have studied comments carefully and have made correction which we hope meet with approval. Then this manuscript has been improved by a professional editing company. We also respond point by point to the reviewer’s comments, along with the clear manuscript as well as a version with track changes as supplementary file. The main corrections in the paper and the responds to the reviewer’s comments are as flowing:

Reviewer #1:

Comment 1: This part should contain brief information of the introduction such as the importance of this crop and disease.

Response: Thanks. Considering the Reviewer’s suggestion, in the part of “Abstract”, the introduction of the Saposhnikovia divaricata and root rot disease have been supplemented at line 10-13.

Comment 2: This section lacks citations-references, only 10 different references is not acceptable for an introduction in a scientific publication.

Response: Thanks. We reworded this part and cited more reference.

Comment 3: The authors should include statistical parameters from analysis of variance (ANOVA) such as F value, dr or p value.

Response: Thanks. But F value, dr or p value are the process in the analysis of variance, and they rarely including statistical parameters like as articles as following,

  • Daroodi, Z. Taheri, P. and Tarighi, S. Direct antagonistic activity and tomato resistance induction of the endophytic fungus Acrophialophora jodhpurensis against Rhizoctonia solani, Biol. Control. 2021, 160, 104696.
  • Yan Y. Mao Q. Wang Y. et al. Trichoderma harzianum induces resistance to root-knot nematodes by increasing secondary metabolite synthesis and defense-related enzyme activity in Solanum lycopersicum Biol. Control. 2021, 158, 104609.

Comment 4: Table 2: A table only with two columns is not acceptable in a scientific publication, authors should change by a figure.

Response: thanks. We agreed and draw the figure.

We hope that all these changes fulfil the requirements to make the manuscript acceptable for publication in Agronomy.

Thank you very much again.

Regards,

Zhongming

Reviewer 2 Report

This manuscript is good and describes important studies from the biological control point of view. I appreciate the work and have few comments to improve the manuscript.

Title

In my opinion, the title is not stylistically written.

Materials and Methods

P2, L71: Please cite any previous paper that describes the isolation procedure or write it briefly in this manuscript.

P2, L82: I would write either the names of all the tested pathogens or none.

P2, L90-92: Please add a verb in the sentence.                       

P4, L175: Remove “and growth promotion”. You don’t describe the method of growth promotion studies in this section.

P4, L192: Remove S. divaricate

P5, L204: In Statistical analysis, I suggest adding one more time information about the number of replications.

Results

Why do you use the term ‘antibiotic activity? Please explain that in the manuscript.

P5, L226: Please explain as compared to what, the results were significantly different.

Figure 5: Remove  “This is a figure.”

P9, L321: Change “S. divaricate” to “S. divaricate” *the same P10, L340), L322: Remove “of S. divaricate

Discussion

P10, L358: Remove “and a new record”

P11, L368: In my opinion, the information that antagonistic fungi may control the disease by induction of defense responses and resistance in plants, should be added.

P11, L373: Antibacterial or fungistatic?

Author Response

Thank you for your letter and for the reviewers’ comments concerning our manuscript entitled “New observation in biocontrol activity of Acrophialophora jodhpurensis against Fusarium equiseti on Saposhnikovia divaricata and as a plant growth promoter” (Manuscript ID: agronomy-1919858). Those comments are all valuable and very helpful for revising and improving our paper, as well as the important guiding significance to our researches. We have studied comments carefully and have made correction which we hope meet with approval. Then this manuscript has been improved by a professional editing company. We also respond point by point to the reviewer’s comments, along with the clear manuscript as well as a version with track changes as supplementary file. The main corrections in the paper and the responds to the reviewer’s comments are as flowing:

Reviewer #2:

Comment 1: In my opinion, the title is not stylistically written.

Response: We reworded the title that is “Effect of rhizospheric fungus on biological control of root rot (Fusarium equiseti) disease of Saposhnikovia divaricata ”.

Comment 2: P2, L71: Please cite any previous paper that describes the isolation procedure or write it briefly in this manuscript.

Response: The fungi isolation has been described in the manuscript (line 86-87).

Comment 3: P2, L82: I would write either the names of all the tested pathogens or none.

Response: Thanks. We wrote all the tested pathogens (Line 100-102).

Comment 4: P2, L90-92: Please add a verb in the sentence.

Response: We have been corrected as “Observation of culture characteristics: MR-57 isolate inoculated in the Cornmeal Agar (CMA), Malt Extract Agar (MEA), Oatmeal Agar (OA) and Plate Count Agar (PCA) medium plates in three-point fashion under aseptic conditions with three replications each time, respectively. The plates were incubated at 25 ℃ in the dark for 10 days” (Line 110-113).

Comment 5: P4, L175: Remove “and growth promotion”. You don’t describe the method of growth promotion studies in this section.

Response: Thanks. The“and growth promotion” have been deleted.

Comment 6: P4, L192: Remove S. divaricate

Response: Thanks. the “S. divaricata” have been deleted.

Comment 7: P5, L204: In Statistical analysis, I suggest adding one more time information about the number of replications.

Response: Considering the Reviewer’s suggestion, the information about number of replications have been described at line 233-234.

Comment 8: Why do you use the term ‘antibiotic activity? Please explain that in the manuscript.

Response: “antibiotic activity” have been corrected as “antagonistic activity” in the manuscript.

Comment 9: P5, L226: Please explain as compared to what, the results were significantly different.

Response: Thanks. The “significantly” have been corrected as “strongly”.

Comment 10: Figure 5: Remove “This is a figure.”

Response: Thanks. We deleted it.

Comment 11: P9, L321: Change “S. divaricate” to “S. divaricate” *the same P10, L340), L322: Remove “of S. divaricate”

Response: This part have been corrected.

Comment 12: P10, L358: Remove “and a new record”

Response: “and a new record” have been deleted in the manuscript, and described as “and it is the first time that this fungal species was isolated in China.”

Comment 13: P11, L368: In my opinion, the information that antagonistic fungi may control the disease by induction of defense responses and resistance in plants, should be added.

Response: The information about plant resistance induction have been described.

Comment 14: P11, L373: Antibacterial or fungistatic?

Response: Line 393, “extracellular antibacterial substances” have been corrected as “extracellular fungistatic substances”.

We hope that all these changes fulfil the requirements to make the manuscript acceptable for publication in Agronomy.

Thank you very much again.

Regards,

Zhongming

Reviewer 3 Report

I understand that the authors report the potential as a biological control agent of the isolate MR-57 (presumably Acr. jodhpurensis) against the damage caused by F. equiseti in S. divaricata. Additionally, they claim to be the first to report Acr. jodhpurensis in China.

Main remarks

-Origin of microorganisms. Acr. jodhpurensis is also an endophyte and in MM (2.1) the isolation procedure is not described, I am not sure if it was isolated from root hairs or from the rhizosphere. On the other hand, the origin and identity of the other fungi are not specified.

- Identity of isolate MR-57. Acr. jodhpurensis is similar to Acr. teleoafricana, treated as sister species according to reference 22. Morphologically, the length of the ascospores of MR-57 cannot be differentiated from the range of variation between both species (according to reference 22, (11–12.5 × 6–7 μm vs 13 –14.5 × 6–7 μm) In this situation at least one other gene is needed to establish identity (perhaps tub2 or rpb2), as the morphology data presented (including Figure 2) are not convincing to me to differentiate between the two species, not even others within the genus.

-Pathology. I do not see the significance of F. equiseti in the agronomy of S. divaricata, there are no additional references on the symptoms in the field (line 40), therefore I am not sure that the symptoms in the greenhouse are the ones required to observe (lines 326 and 327 ). No papers on the phytopathology of this plant-pathogen interaction are cited in the introduction.

-Endophytic nature of Acr. jodhpurensis. I am confused after reading the manuscript regarding the authors' criteria regarding the habitat of this species. It was isolated from the rhizosphere, but is also known as an endophyte. Neither the introduction nor the discussion refers to this duality. The experiments described ignore the possibility of infection by being in soil.

Details

Title. "New observation..." does not contribute to an informative title.

MM

- Line 150. The final concentration was 10x3, and not 10x6 (line 147).

-"plastic greenhouse"? What does it mean, temperature and lighting control, a plastic foil under natural conditions? Be specific.

- 2.7. Here the method should be explained, and mention that 10 generations of subculture were done to achieve stability of the "mutant" MRRif-57. Reference 18 only indicates a method of identifying resistant from natural populations. Having even a "mutant", how did it recover from non-sterile soil? and how did the mutant differ from the microbial community?

-2.8 and 2.9 it is not clear if they are two separate experiments, or one. Indicators for growth promotion are not described in 2.8. Think about joining both items. Explain what is disease index and what is protection rate. Describe the origin of the soil "filled with soil in which S. divaricata was grown" and where all the seedlings of S. divaricata used in the study came from. I understand that the study was carried out in a greenhouse, not in the field, to clarify what should be understood by "conventional agricultural management".

RESULTS

- I recommend combining tables 1 and 2, and converting table 1 into text since it does not contribute anything to the discussion made by the authors. I consider that the titles should be more informative and use part of the information contained at the foot of the tables.

- Fig. 2 of the manuscript does not seem similar to Fig. 13 of reference 22, (a), (b) and (c) are not informative enough to distinguish the structures to which it refers. I recommend deleting Fig. 2. Fig. 1 I think is necessary, however the colonies do not appear fresh to show the distinctive color or texture (CMA, MEA and OA).

-Fig. 4 c-e, I am not convinced that what is indicated is an artifact of the manipulation of the mycelium during the preparation of the sample. The structure that is being pointed out in each of the images is not described.

-3.6. what is SWRif-34? the fig. 5 indicates MRRif-57 in the title and y on the y-axis points to MRRfi-16. The last line of the title is unnecessary (x-axis).

-Table 4 and 5. It would help the reader a lot if the treatments were named in a way that refers to the active ingredients. It draws my attention that there are no significant differences A and B, and there are between C and D. Explain the origin of the incidence values ​​and %, in addition to the n (also in table 5). Plant is understood to be everything, I think it would be more obvious to the reader to use the terms stem/root.

DISCUSSION

- line 373-74 I think it's a lot of speculation for the results shown.

-Line 381-83 "antibiotic labelling" is something desirable to monitor the pathogen without having to transform it by genetic engineering, however I have doubts because we do not know its effects on its capacity for infection as an endophyte or as a resistance inducer if it only inhabits in the rhizosphere.

Author Response

Thank you for your letter and for the reviewers’ comments concerning our manuscript entitled “New observation in biocontrol activity of Acrophialophora jodhpurensis against Fusarium equiseti on Saposhnikovia divaricata and as a plant growth promoter” (Manuscript ID: agronomy-1919858). Those comments are all valuable and very helpful for revising and improving our paper, as well as the important guiding significance to our researches. We have studied comments carefully and have made correction which we hope meet with approval. Then this manuscript has been improved by a professional editing company. We also respond point by point to the reviewer’s comments, along with the clear manuscript as well as a version with track changes as supplementary file. The main corrections in the paper and the responds to the reviewer’s comments are as flowing:

Reviewer #3:

Comment 1: Origin of microorganisms. Acr. jodhpurensis is also an endophyte and in MM (2.1) the isolation procedure is not described, I am not sure if it was isolated from root hairs or from the rhizosphere. On the other hand, the origin and identity of the other fungi are not specified.

Response: Considering the Reviewer’s suggestion, the fungi isolation and origin and identity of pathogenic fungi have been supplemented at line 86-87, 89-96.

Comment 2: Identity of isolate MR-57. Acr. jodhpurensis is similar to Acr. teleoafricana, treated as sister species according to reference 22. Morphologically, the length of the ascospores of MR-57 cannot be differentiated from the range of variation between both species (according to reference 22, (11–12.5 × 6–7 μm vs 13 –14.5 × 6–7 μm) In this situation at least one other gene is needed to establish identity (perhaps tub2 or rpb2), as the morphology data presented (including Figure 2) are not convincing to me to differentiate between the two species, not even others within the genus.

Response: Thanks. We agreed that the length of the ascospores of MR-57 cannot be differentiated from the range of variation between Acr. jodhpurensis is similar to Acr. teleoafricana based on the analysis of morphological traits. But we did rDNA internal transcribed spacers (ITS) analysis and rebuild phylogenetic tree (Fig. 3) including MR-57 and Acr. teleoafricana. The result showed that the strain MR-57 showed the highest sequence homology (100%) with Acrophialophora jodhpurensis (MN889997.1) were  in the same clade. Based on phylogenetic tree construction, strain MR-57 and Acr. teleoafricana were in the different clade. So, combined morphological traits and ITS analysis, we confirmed that strain MR-57 is Acrophialophora iodhpurensis.

Comment 3: Pathology. I do not see the significance of F. equiseti in the agronomy of S. divaricata, there are no additional references on the symptoms in the field (line 40), therefore I am not sure that the symptoms in the greenhouse are the ones required to observe (lines 326 and 327). No papers on the phytopathology of this plant-pathogen interaction are cited in the introduction.

Response: The symptoms of S. divaricata root rot have been described that after the disease outbreak, symptoms of root rot are brown rotting spots of main roots, yellowing of leaves, and eventually wilt and metabolic failure of the aboveground plant of S. divaricata (line 50-52).

Comment 4: Endophytic nature of Acr. jodhpurensis. I am confused after reading the manuscript regarding the authors' criteria regarding the habitat of this species. It was isolated from the rhizosphere, but is also known as an endophyte. Neither the introduction nor the discussion refers to this duality. The experiments described ignore the possibility of infection by being in soil.

Response: This part have been supplemented in the “discussion” as following:

It has been reported that Acrophialophora can prevent and control plant diseases. Daoodi et al. [37] reported the endophytic fungus Acr. jodhpurensis isolated from tomato roots as a potential biocontrol source against Rhizoctonia solani. In this study, Acr. jodhpurensis isolated from the rhizospheric soil of S. divaricata, this suggests that Acr. jodhpurensis maybe a member of the soil saprophytic group of fungi (first isolated from rabbit dung, have saprophytic character), meanwhile, this species have the strongly endophytic ability.

Comment 5: "New observation..." does not contribute to an informative title.

Response: The title was revised as “Effect of rhizospheric fungus on biological control of root rot (Fusarium equiseti) disease of Saposhnikovia divaricata ”.

Comment 5: - Line 150. The final concentration was 10x3, and not 10x6 (line 147).

Response: This part has been confirmed that the final concentration was 106.

Comment 6: -"plastic greenhouse"? What does it mean, temperature and lighting control, a plastic foil under natural conditions? Be specific.

Response: Thanks. We reworded this sentence that in this assay, all treatments were performed in the phytotron over 35 days (16 h sunlight period at 28 °C, 8 h dark period at 16 °C) .

Comment 7: 2.7. Here the method should be explained, and mention that 10 generations of subculture were done to achieve stability of the "mutant" MRRif-57. Reference 18 only indicates a method of identifying resistant from natural populations. Having even a "mutant", how did it recover from non-sterile soil? and how did the mutant differ from the microbial community?

Response: In this part, the rifampicin-resistant (Rift) mutant (strain MRRif-57) could grow stably on PDA plates containing a concentration of 400 μg·ml-1 rifampicin, and the other soil microorganism could not grow on this PDA plates. By this way, strain MRRif-57 could be isolated from the soil, and this mothed was confirmed according to repeatedly experimental verifications.

Comment 7: 2.8 and 2.9 it is not clear if they are two separate experiments, or one. Indicators for growth promotion are not described in 2.8. Think about joining both items. Explain what is disease index and what is protection rate. Describe the origin of the soil "filled with soil in which S. divaricata was grown" and where all the seedlings of S. divaricata used in the study came from. I understand that the study was carried out in a greenhouse, not in the field, to clarify what should be understood by "conventional agricultural management".

Response: Part 2.8 and 2.9 are two separate experiments, and title of part 2.8 have been corrected. Disease index and control efficacy have been supplemented at line 195-202. In this study, the 1-year-old seedlings of S. divaricata were collected from medicinal plant garden of Jilin agricultural university. The part of "conventional agricultural management" have been corrected or deleted.

Comment 8: I recommend combining tables 1 and 2, and converting table 1 into text since it does not contribute anything to the discussion made by the authors. I consider that the titles should be more informative and use part of the information contained at the foot of the tables.

Response: The table 1 may help to reveal the antagonistic activities of the other fungal isolates intuitively and clearly. so we suggested that table 1 could be remained as supporting information (Tab. S1).

Comment 9: - Fig. 2 of the manuscript does not seem similar to Fig. 13 of reference 22, (a), (b) and (c) are not informative enough to distinguish the structures to which it refers. I recommend deleting Fig. 2. Fig. 1 I think is necessary, however the colonies do not appear fresh to show the distinctive color or texture (CMA, MEA and OA).

Response: The Figure 3 (the original Figure 2 has been change to Figure 3) may help to reveal the microscopic features of MR-57, so it is recommended that the Fig. 3 should be remained in the manuscript; and due to the zoom and camera angle, Fig. 3 can be 100% similar to Fig. 13 of reference 22, but Fig. 3a are similar to Fig.13d, the only difference is that Fig. 3a is one ascomata and Fig.13d has several ascomata. moreover, the colony cultured on MEA and PCA in the Fig. 3 were very similar to these of Fig. 12 of reference 22. So we suggested that Fig. 3 should be remained.

In the study, the culture characteristics of MR-57 was observed every day, but we only reserved the pictures witch incubated for 10 days. Unfortunately, the photo effect of these photographs was not ideal, so we can't provide the other evidence of photos. Once again, we feel really sorry about this.

Comment 10: -Fig. 4 c-e, I am not convinced that what is indicated is an artifact of the manipulation of the mycelium during the preparation of the sample. The structure that is being pointed out in each of the images is not described.

Response: The Figure 5 (The original figure 4 has been change to figure 5) may help to reveal the microscopic features of MR-57, so it is recommended that the Figure 4 should be remained in the manuscript. Considering the Reviewer’s suggestion, we reworded this part that microscopic observation of the mycelial morphology of F. equiseti showed that the healthy mycelia of F. equiseti were observed with the smooth exterior surface (Figure 5c). However, the surface of inhibited mycelia was rough and uneven, increased with irregular extension, and the branching ends were enlarged, with condensed inclusions and uneven distribution. The inhibited mycelia were distorted, folding, constricted, broken, and fractured (Figure 5d, e) (line 305-310).

Comment 11: -3.6. what is SWRif-34? the fig. 5 indicates MRRif-57 in the title and y on the y-axis points to MRRfi-16. The last line of the title is unnecessary (x-axis).

Response: Thanks. We revised this part as following.

The strain MRRif-57 could still grow stably on PDA plates containing a concentration of 400 μg·ml-1 rifampicin after 10 generations of subculture without a noticeable change in the morphology. The strain MR-57 and MRRif-57 were similarly effective against F. equiseti, the inhibition rate of MRRif-57 was up to 60%. The results indicate that strain MRRif-57 was genetically stable and still maintained high inhibition activity against F. equiseti (line 330-334).

Comment 13: -Table 4 and 5. It would help the reader a lot if the treatments were named in a way that refers to the active ingredients. It draws my attention that there are no significant differences A and B, and there are between C and D. Explain the origin of the incidence values and %, in addition to the n (also in table 5). Plant is understood to be everything, I think it would be more obvious to the reader to use the terms stem/root.

Response: Considering the Reviewer’s suggestion, the part of 3.7 and table 4 have been corrected at line 346-360. The incidence values and % of disease index and control efficacy have been explained in the part of 2.8. the part of 3.8 and table 5 have been corrected at line 361-374.

Comment 14: -Line 381-83 "antibiotic labelling" is something desirable to monitor the pathogen without having to transform it by genetic engineering, however I have doubts because we do not know its effects on its capacity for infection as an endophyte or as a resistance inducer if it only inhabits in the rhizosphere.

Response: In this study, we confirmed the strain MR-57 could grow and propagate stably in the soil by antibiotic labelling method, and in further studies we will explore systematically the capacity for infection as an endophyte or soil saprophytic fungus by genetic engineering methods.

We hope that all these changes fulfil the requirements to make the manuscript acceptable for publication in Agronomy.

Thank you very much again.

Regards,

Zhongming

Round 2

Reviewer 1 Report

Manuscript ID: agronomy-1919858

Manuscript comments:

After a second revision round, the authors did not carry out my suggestions as I indicated in the first round. Many minor mistakes have not been improved. In addition, the authors did not address major suggestions such as including more references in the introduction section and the statistical parameters (F value, dr and p value) per each result. This last indication is the most important (major revision) and the authors did not include these parameters. Even, the authors indicated two-way ANOVA when all the results are showed as one-way ANOVA. All these mistakes make a not reliable and trustable work. The new figure 1 is small and I cannot read the words, the quality of it is not acceptable for a publication.

Finally, the manuscript did not improve from the previous revision and should not be acceptable for a journal as Agronomy.

Author Response

Dear Reviewer:

Thank you for your letter and for the reviewers’ comments concerning our manuscript entitled “New observation in biocontrol activity of Acrophialophora jodhpurensis against Fusarium equiseti on Saposhnikovia divaricata and as a plant growth promoter” (Manuscript ID: agronomy-1919858). Those comments are all valuable and very helpful for revising and improving our paper, as well as the important guiding significance to our researches. We have studied comments carefully and have made correction which we hope meet with approval. Revised portion are marked in red in the paper. The main corrections in the paper and the responds to the reviewer’s comments are as flowing:

Reviewer #1:

After a second revision round, the authors did not carry out my suggestions as I indicated in the first round. Many minor mistakes have not been improved. In addition, the authors did not address major suggestions such as including more references in the introduction section and the statistical parameters (F value, dr and p value) per each result. This last indication is the most important (major revision) and the authors did not include these parameters. Even, the authors indicated two-way ANOVA when all the results are showed as one-way ANOVA. All these mistakes make a not reliable and trustable work. The new figure 1 is small and I cannot read the words, the quality of it is not acceptable for a publication.

Response:

Many thanks for the reviewer’s comments.

First: We had cited more reference in the introduction and there are 18 references in the revised manuscript.

Second: The statistical parameters including F and p value have been supplemented in the table of section 3.5, 3.7 and 3.8.

Third: we are very sorry about the description of two-way ANOVA in the section of statistical analysis. Actually, all results were analyzed by one-way ANOVA, not two-way ANOVA. Thanks for your suggestion. So we have reworded the statistical analysis of this part that the results were analyzed by one-way ANOVA with 95% confidence intervals for Dun-can's DMRT.

Fourth: Figure 1 have been redrawn a clear one.

Fifth: all the minor mistakes in the manuscript had been checked and revised.

We hope that all these changes fulfil the requirements to make the manuscript acceptable for publication in Agronomy.

Thank you very much again.

Regards,

Zhongming

Reviewer 3 Report

Dear Author,

Thank you for the second version of the manuscript. It improved substantially.

Some final minor suggestions that are not invalidating for publication. 

1) Section 2.4 was a previous step to 2.5, it would simplify the reading if both are joined, similar to how experiment 2.2 was described.

2) i-Fig. 1 does not easily show the names of the pathogens (image resolution??). ii-The title could be more explanatory, in order to differentiate it from other results presented. Something like "... in dual culture assays on PDA"???.

3) Check the wording of the first sentence of the Conclusion, the verb "was" is missing???

Author Response

Dear Reviewers:

Thank you for your letter and for the reviewers’ comments concerning our manuscript entitled “New observation in biocontrol activity of Acrophialophora jodhpurensis against Fusarium equiseti on Saposhnikovia divaricata and as a plant growth promoter” (Manuscript ID: agronomy-1919858). Those comments are all valuable and very helpful for revising and improving our paper, as well as the important guiding significance to our researches. We have studied comments carefully and have made correction which we hope meet with approval. Revised portion are marked in red in the paper. The main corrections in the paper and the responds to the reviewer’s comments are as flowing:

Reviewer #3:

Comment 1: Section 2.4 was a previous step to 2.5, it would simplify the reading if both are joined, similar to how experiment 2.2 was described.

Response: Considering the reviewer’s suggestion, the Section 2.4 and 2.5 have been modified at line 141-154.

Comment 2: does not easily show the names of the pathogens (image resolution??). ii-The title could be more explanatory, in order to differentiate it from other results presented. Something like "... in dual culture assays on PDA"???.

Response: Thanks for your suggestion. The figure 1 have been modified.

Comment 3: Check the wording of the first sentence of the Conclusion, the verb "was" is missing???

Response: Considering the Reviewer’s suggestion, this part have been modified at line 448-449.

We hope that all these changes fulfil the requirements to make the manuscript acceptable for publication in Agronomy.

Thank you very much again.

Regards,

Zhongming
